# Unveiling Ancestral Sustainability: A Comprehensive Study of Economic, Environmental, and Social Factors in Potato and Quinoa Cultivation in the Highland Aynokas of Puno, Peru

Fredy Calizaya [1,2,*], Luz Gómez [3], Jorge Zegarra [4], Melvin Pozo [5], Carmen Mindani [6], Cirilo Caira [7] and Elmer Calizaya [8]

1   Facultad de Ciencias Agrarias, Escuela Profesional de Ingeniería Agronómica, Universidad Nacional del Altiplano, Puno 21001, Peru
2   Programa de Doctorado en Agricultura Sustentable (PDAS), Universidad Nacional Agraria La Molina, Ave. La Molina S.N., Lima 15012, Peru
3   Departamento de Fitotecnia, Facultad de Agronomía, Universidad Nacional Agraria La Molina, Ave. La Molina S.N., Lima 12056, Peru; luzgomez@lamolina.edu.pe
4   Escuela Profesional de Ingeniería Agronómica y Agrícola, Universidad Católica de Santa María Arequipa, Urbanización San José s/n Umacollo, Arequipa 04445, Peru; jzegarraf@ucsm.edu.pe
5   Facultad de Ciencias Agrarias Escuela Profesional de Agronomía, Universidad Nacional de Huancavelica, Paturpampa 09001, Peru; melvin.pozo@unh.edu.pe
6   Facultad de Industrias Alimentarias, Universidad Nacional Agraria La Molina, Av. La Molina S.N., Lima 12056, Peru; 20190596@lamolina.edu.pe
7   Facultad de Ingeniería, Escuela Profesional Ingeniería Forestal y Ambiental, Universidad Nacional de Jaén, Jaen 06800, Peru; cirilo.ccaira@unj.edu.pe
8   Facultad de Ciencias Agrarias, Escuela Profesional de Ingeniería Topográfica y Agrimensura, Universidad Nacional del Altiplano, Puno 21001, Peru; ecalizaya@unap.edu.pe
*   Correspondence: fcalizaya@unap.edu.pe; Tel.: +51-951671308

**Abstract:** Centuries of cultivation in the Highland Aynoka of Puno, Peru, have endowed indigenous crops such as potato and quinoa with rich cultural and nutritional value deeply ingrained in local traditions. This study meticulously evaluates their economic viability, environmental implications, and cultural importance by employing a mixed-methods research approach involving surveys, interviews, and observations. The outcome reveals that while the Economic Sustainability Index (EKI) moderately supports potato and quinoa production sustainability, with a value of 2.98, it falls short of significant impact. Conversely, the Environmental Sustainability Index (ESI) and the Social Sustainability Index (SSI) exhibit moderate levels of sustainability, recording values of 4.04 and 3.38 for ESI and SSI, respectively. These crops demonstrate acceptable economic feasibility, marked by consistent sales, income generation, and manageable production expenses. The findings underscore the urgency of endorsing sustainable farming methods to safeguard cultural heritage, boost market prospects, and fortify regional ecological robustness. Rooted in ancestral sustainability, potato and quinoa cultivation is a cornerstone in local food systems. Recognizing the cultural, economic, and environmental significance inherent to these crops, efforts can be channeled towards nurturing sustainable agricultural systems that uphold community well-being, conserve biodiversity, and facilitate cultural resilience in Puno's Highland Aynoka.

**Keywords:** sustainability; ancestral; Aynoka; survey; indicators; potato and quinoa; highland

## 1. Introduction

The "Aynoka" is an Aymara word used to refer to an ancestral farming system that is characterized by the sustainable management and use of land, with crop rotation and plant diversity, in large communal areas that ensure the quantity and quality of food for the family and the community [1]. The Aynoka exist because they are part of an organized community agricultural system, where members respect communal decisions by accepting

crops, cultivation technologies, and the sharing of the harvest, ensuring the well-being of community members [2,3]. Aynoka are small plots of land cultivated by families or local communities, generally located near their homes. These agricultural spaces can have different purposes, such as growing food for self-consumption, generating additional income by selling agricultural products, or preserving traditional farming practices [4].

The most commonly used rotation system in Aynoka is potato, quinoa, barley, or oats, with a fallow period lasting approximately 8 to 10 years. It includes a crop cycle for each species over the entire cultivated area of the community of many hectares. It takes 3 to 4 years for the cultivated species and a rest period of 6 to 7 years for the plots covered with natural pastures [5]. Any change in the crop rotation sequence requires community consensus [6]. The cycle includes crops native to Peru, such as potato and quinoa, and crops introduced more than 500 years ago, such as barley and oats. It is essential to highlight that biodiversity is maintained in each species cultivated because farmers plant wide varieties and ecotypes so that Aynoka is a germplasm bank where natural crossings or gene flows occur, generating more extraordinary variability year after year, which is further enriched by natural mutations and the contribution of genes from wild relatives growing naturally in the vicinity [7]. Food security is based on the presence of many different genotypes and their adaptation to the area, and the soils remain healthy and prosperous. Its versatile nature and inherent adaptability have transformed it into a crucial element of food security, safeguarding against crop failures and external disruptions [8]. After all, the fallow period restores fertility, especially with animal grazing, enhancing the soil with manure and facilitating the reemergence of native species such as clover; these factors collectively ensure the system's robustness against external shocks and ensure the long-term viability of food production [9]. It serves as a period of isolation, decreasing soil contamination from fungi, nematodes, and insects and supporting the proliferation of indigenous flora and fauna.

This ancestral cultivation system was applied for thousands of years in the Bolivian Peruvian altiplano and persists today. This historical continuity has established a sense of cultural identity and pride, perpetuating the practices through storytelling and communal traditions [10]. However, in recent years, the rotation sequence of traditional crops such as quinoa and potato has been modified, especially in areas with access to technical irrigation [11]. In the western half of South America, the potato is crucial in indigenous history, culture, and selfhood. Its cultivation and consumption have shaped traditional practices, culinary traditions, and spiritual beliefs, while its genetic diversity has contributed to food security and resilience in the region [12–14]. Converging relationships between people, plants, and the environment have created a vibrant and diverse potato landscape in the Andean region [15]. Native to the Andean highlands, the potato has been cultivated for over 7000 years, evolving alongside human needs and cultural preferences. The region boasts unparalleled genetic diversity, with thousands of potato varieties adapted to different altitudes and climates [16]. Native potato diversity is vital in enhancing community resilience and serving as a critical source of food security [17].

Quinoa has enormous genetic variability and adaptability to different climatic conditions and has a balanced nutrient content, providing an adequate combination of carbohydrates, proteins, and fats. This characteristic makes it a complete and beneficial food for maintaining a balanced and healthy diet [18]. Quinoa and other indigenous foods can play a valuable role in the fight against hunger and malnutrition, especially in climate change. Their resilience, nutritional value, and ability to diversify food systems make them essential tools for strengthening food security in low-income countries dependent on agriculture and facing limited inputs (Food and Agriculture Organization of the United Nations) [19].

It is important to note that variations in agroecological conditions can result in differences in the nutritional composition of quinoa between different regions and varieties. These variations can be used strategically to select quinoa varieties and farming practices that maximize their nutritional content in agroecological contexts [20,21]. Sustainable agriculture is developing globally and is constantly evolving in three key areas: economic,

environmental, and social, encompassing five levels, including the field, the farm, the local community, and the national and international levels [22] The economic capacity and commitment to the sustainability of farms directly impact a country's food quality and the environment [23]. Prioritizing profits over the environment and society results in dissatisfaction with conventional farming, promoting the switch to sustainable methods [24]. Evaluating social facets in agri-environmental initiatives, specifically sociocultural factors affecting participation quality and social welfare outcomes, is constrained [25]. The demand for certain species and the need to increase yields have determined the inclusion of techniques from the conventional agricultural system. These changes are already occurring permanently and could be affecting the sustainability of the Aynoka, so it is crucial to characterize the current system and identify the limiting factors that make it possible to propose research that contributes to the sustainability of the system and food security [26]. Nevertheless, the Aynoka system's resilience faces challenges in the modern era. Changes in crop rotation due to technical irrigation and external market demands raise questions about the system's sustainability [27]. This study endeavors to generate insights that can guide strategies for enhancing the sustainability and longevity of potato and quinoa cultivation within the Aynoka system. Finally, the traditional crop rotation system in Aynoka persists due to the convergence of historical resilience, ecological harmony, and cultural cohesion. Culturing the potato and quinoa plant in the Andean region magnifies the intricate dynamics between human practices, genetic diversity, and food security. In an era marked by the urgency of global food security and sustainable development, this study is pivotal in preserving an ancient agricultural heritage while adapting it to the complexities of the modern world [28,29].

The research aims to comprehensively understand the cultivation practices, sustainability indicators, and their impact on potato and quinoa crops within the Aynoka system. By characterizing and assessing these crops, the study seeks to contribute to developing strategies that enhance the sustainability and long-term viability of potato and quinoa production while considering the unique context of the Palermo Rio Salado Community.

## 2. Materials and Methods

### 2.1. Study Site

The study was implemented in the Palermo Rio Salado community in the province of Chucuito Juli, within the Puno Region of Peru. The community is 3831 m above sea level, with specific geographical coordinates of Longitude: 69°30′10.55″ W and Latitude: 16°19′43.39″ S (Figure 1). Palermo Rio Salado is home to approximately 300 farmers who primarily cultivate quinoa, potatoes, and other agricultural products. It should be noted that while the community consists of around 300 farmers, only 130 are officially registered and reside within the community. Despite this, all landowners within the community uphold the ancestral practice of the Aynoka system. The climate in Juli, as classified by the Köppen–Geiger climate classification, is considered and characterized as a tundra climate with high mountain influences. The average annual rainfall in Juli is recorded at 801 mm, with July being the driest month, receiving only 5 mm of precipitation, and January being the wettest month, with an average of 173 mm of rainfall. The average annual temperature in the region is 8.3 °C, with December being the warmest month, averaging 9.6 °C, and June being the coldest month, with an average temperature of 6.2 °C [30]. Hydrographically, the community is situated within the micro-watershed of the Salado River. It is important to note that the Salado River exhibits a high pH level, exceeding 8, which poses limitations for irrigated agriculture and diminishes the availability of drinking water.

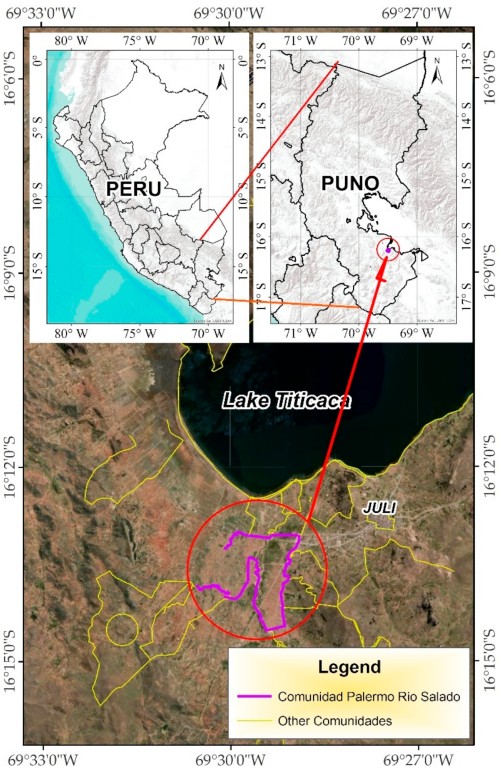

**Figure 1.** Location of the Palermo Rio Salado community study area in the Puno Andes.

Consequently, the community must seek water sources from neighboring communities, which can potentially give rise to conflicts due to the significance of this shared water resource. These geographic and environmental conditions shape the agricultural landscape and practices within the community of Palermo Rio Salado, influencing the strategies employed by farmers to overcome challenges related to water scarcity, soil management, and climatic conditions. Understanding the specific context of the community provides valuable insights into the unique circumstances and considerations that shape the sustainability of potato and quinoa cultivation in this region.

### 2.2. Population and Sample

From the population of 130 registered producers of quinoa and potato, an unrestricted random sample of (n = 55) was obtained using the survey design and analysis formula [31,32]. According to the model, farmers were selected by random sampling in the community of Palermo Rio Salado [33]. Equation (1):

$$n = \frac{N * Z_\propto^2 * p * q}{E^2 * (N-1) + Z_\propto^2 * p * q} \tag{1}$$

where: n = Sample size; $N$ = Total population 130 (papa and quinoa producers in the community from two campaigns); $Z$ = 95% confidence level (1.96); $p$ = Probability in favor 50%; $q$ = Probability against 50%; $e$ = Sample error 10%.

### 2.3. Variables Assessed

Variables of social (housing, education, family composition, and levels of social integration), economic (land tenure, labor sources, productivity, cost of production, sales price, and added value of primary products), and environmental (pest control, pollution, rotation systems, soil slope management, genetic diversity, among the most important) were identified and adapted (Table 1).

**Table 1.** Characteristics of the economic, environmental, and social variables of the Palermo Rio Salado community.

| Economic | Environmental | Social |
|---|---|---|
| A. Profitability | A. Soil life conservation | A. Satisfaction of basic needs |
| A1. Cultivated surface | A1. Crop rotation | A1. Housing type |
| A2. Productivity | A2. Crop diversification | A2. Level of education |
| A3. Pest incidence | A3. Organic matter incorporation | A3. Access to health insurance |
| B. Economic income | A4. Land preparation | A4. Basic services |
| B1. Monthly net income | B. Risk of erosion | B. Acceptability of production systems |
| C. Economic risk | B1. Predominant slope | B1. Producer satisfaction level |
| C1. Diversification for sales | B2. Vegetation coverage | B2. Production systems |
| C2. Marketing distribution | C. managing biodiversity | C1. Level of social integration |
| C3. Dependence on inputs | C1. Conservation in situ of varieties | D. Technical assistance and training |
| - | C2. Pest/disease management | D1. Level of social assistance and training |
| - | C3. Quality seed production | - |

Figure 2 shows the hierarchical system and the relationship between the agroecosystem functions or dimensions and the principles, criteria, and indicators.

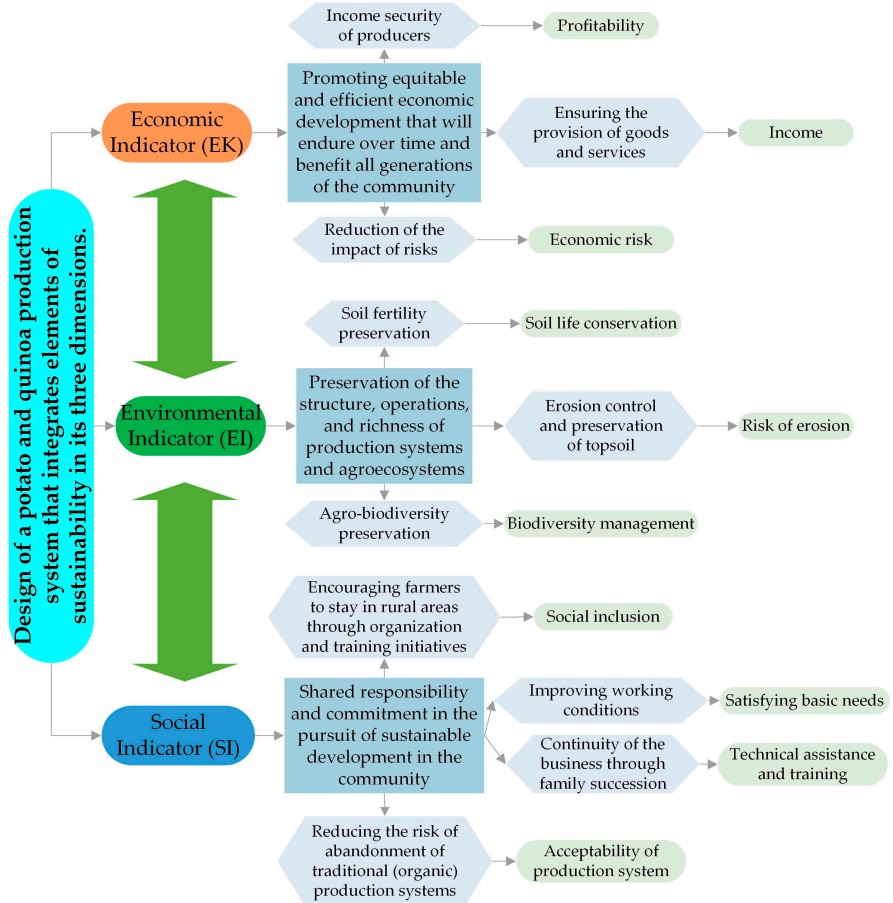

**Figure 2.** Roles, fundamentals, and sustainability criteria for potato and quinoa cultivation.

### 2.4. Sustainability Indicators

For the elaboration and application of sustainability indicators, it was analyzed in depth to explore its scope and limitations. This methodology is considered an essential tool for measuring and assessing the degree of sustainability of productive systems, as it simultaneously allows the evaluation of various aspects such as productivity, environmental, social, and economic sustainability, as well as local culture and temporal dynamics [34]. It

is, therefore, crucial to understand how this methodology can be applied, its benefits, its possible limitations, and how they can be overcome.

Several workshops were held in the farming community of Palermo Rio Salado, where local experts and producers from the study area selected and defined the variables to identify the most critical economic, environmental, and social indicators in potato and quinoa production. Due to COVID-19 restrictions, the meetings were restricted. The methodological sequence was carried out to define the indicators that reflect the current state of potato and quinoa production systems. Figure 3 shows the methodology, which consists of a series of phases that lead to indicators appropriate for assessing the critical points of sustainability of the agroecosystems. The purpose was to create a methodology that is easy to follow and that allows the assessment of those aspects that endanger the sustainability of agricultural systems. The sequence of procedures for elaborating and applying the sustainability indicators is presented below:

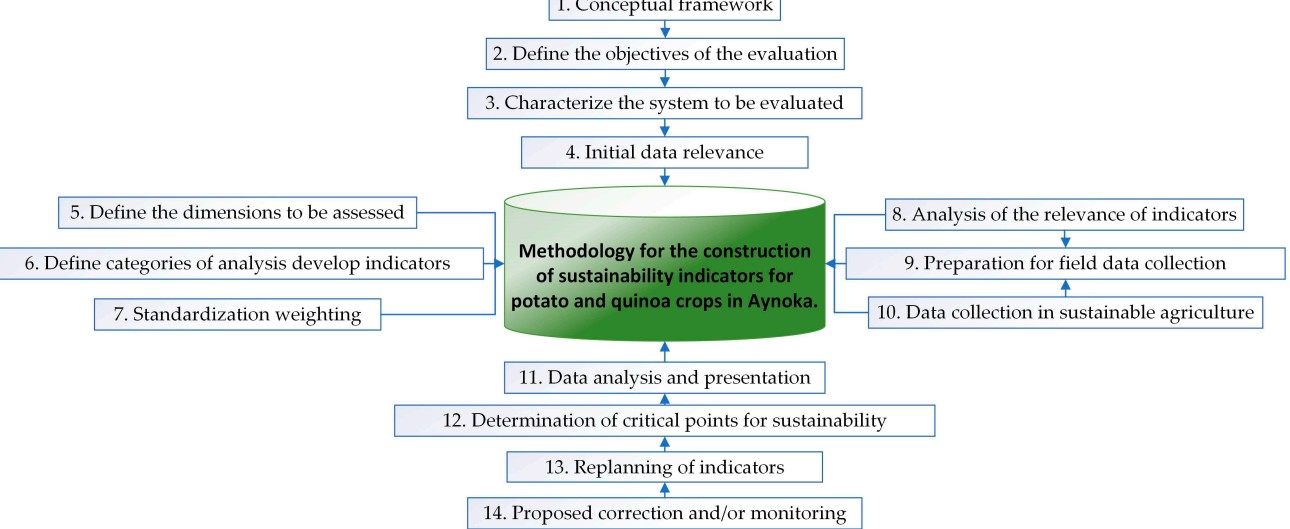

**Figure 3.** Methodology for constructing sustainability indicators for potato and quinoa crops in the Aynoka of Juli—Puno.

### 2.4.1. Value of Economic Indicator

In calculating the economic indicator (EK), the profitability indicator (A) was given importance by assigning a weight of 2. Profitability is crucial for directly assessing the farmer's financial condition. The indicators of net monthly income (B) and economic risk (C) are also considered with a simple weighting. A structure was defined to analyze this economic dimension, including three essential indicators: profitability, net income, and economic risk. These indicators were identified as the most significant and relevant in the specific context of the area under study. Their selection is based on their ability to provide a clear and accurate understanding of the economic situation regarding the income; Equation (2):

$$EK = \frac{\frac{2 * (A1 + A2 + A3)}{3} + \frac{B1}{1} + \frac{(C1 + C2 + C3)}{3}}{4} \tag{2}$$

where: *A*1. Cultivated Surface; *A*2. Productivity; *A*3. Pest incidence; *B*1. Monthly net income; *C*1. Diversification for sales; *C*2. Marketing distribution; *C*3. Dependence on inputs

### 2.4.2. Value of Environmental Indicator

In calculating the environmental indicator (*EI*), it is recognized that the soil life conservation indicator (A) is the most critical. This indicator is directly related to the soil as a primary resource for agricultural production and household food security. Its preservation is fundamental to ensure a sustainable environment and an adequate quality of life.

Within the Palermo Rio Salado farming community, it was determined that the key indicators for measuring environmental aspects include soil life conservation, erosion risk, and biodiversity management. These indicators are fundamental to assessing the qualitative and quantitative characteristics of the ecological setting in this specific community; Equation (3):

$$EI = \frac{\frac{2 * (A1 + A2 + A3 + A4)}{4} + \frac{B1 + B2}{1} + \frac{(C1 + C2 + C3)}{3}}{4} \tag{3}$$

where: *A*1. Crop rotation; *A*2. Crop diversification; *A*3. Organic matter incorporation; *A*4. Land preparation; *B*1. Predominant slope; *B*2. Vegetation coverage; *C*1. Conservation in situ of varieties; *C*2. Pest/disease management; *C*3. Quality seed production.

### 2.4.3. Value of Social Indicator

When weighing the Social Indicator (*SI*), the relevance of the satisfaction of basic needs (A) indicator is highlighted by assigning a duplicate value of 2. This indicator is vital for measuring the family's well-being regarding its access to housing and essential services, which are fundamental elements for a dignified and quality life. In the specific context of the Palermo Rio Salado community, four key indicators were established to assess different aspects relevant to community well-being and development. These indicators cover the measurement of the satisfaction of basic needs, the acceptability of potato and quinoa production systems, social integration, and the valuation of technical assistance and training. Their consideration in the analysis provides a deep and enriching understanding of the social and economic reality in this community; Equation (4):

$$SI = \frac{\frac{2 * (A1 + A2 + A3 + A4)}{4} + \frac{B1 + B2}{2} + \frac{C1}{1} + \frac{D1}{1}}{5} \tag{4}$$

where: *A*1. Housing type; *A*2. Level of education; *A*3. Access to health insurance; *A*4. Essential services, *B*1. Producer satisfaction level; *B*2. Production systems; *C*1. Level of social integration; *D*1. Level of social assistance and training

### *2.5. General Sustainability Index (GSI)*

The selection of the proposed formula for calculating this index is based on its proven effectiveness and relevance in previous research [35]. The calculation of the sustainability index varies according to the context and the criteria established. It generally involves assigning weights to selected indicators and combining them in a formula or equation to obtain a numerical value that reflects the degree of sustainability of the system.

The sustainability index is used as an evaluation and monitoring tool to measure progress toward sustainability and to make informed decisions on policies, practices, and actions to be implemented. Its main objective is to balance human well-being, environmental conservation, and long-term economic viability. All three dimensions are valued equally because, in a correct perspective of sustainability, they are attributed the same relevance and, consequently, given an identical value; Equation (5):

$$GSI = \frac{IK + IA + IS}{3} \tag{5}$$

The criteria established that in the peasant community that practices Aynoka, the index is expected to be higher than 2 as a minimum. It also highlights the importance that none of the indicators corresponding to the three dimensions evaluated should have a value of less than 2 [35]. The sustainability index is a comprehensive measure that assesses the performance of a system in terms of sustainability, considering the different dimensions involved. Its calculation is based on relevant indicators and appropriate weightings to provide an overview of sustainability and guide decision-making toward more sustainable development.

## 2.6. Selection, Evaluation of Indicators for Sustainable Development, and Validation Process

A consensus validation process was used to determine each indicator's relevance, which was done to conduct a comprehensive analysis. A total of 10 indicators and 24 sub-indicators were selected and organized into three dimensions of sustainable development in the methodology proposed [36]. It ensured that all hands were appropriate and relevant to the study. Five response options were established for each sub-indicator, allowing for greater precision in assessing each dimension of sustainable development. This response structure allowed for a wide range of information on each indicator and sub-indicator, which facilitated the identification of strengths and weaknesses in sustainable development in the community. Both hands were assessed on a scale of 1 to 5, with higher points denoting increased sustainability and lower points indicating reduced sustainability (Table 2). Formulae, including weights, were applied to determine the study community's economic, environmental, and social indicators. These weights were assigned according to the degree of importance and each indicator's significance relative to the other indicators assessed.

**Table 2.** Assessment parameters of the General Sustainability Indicator (GSI).

| Scale | Description of Assessment Levels | Sustainability Level |
|:---:|:---|:---:|
| (1) | The very critical or extreme level of the unsustainability of production systems of | Extreme |
| (2) | low or critical level of sustainability of production systems. The production system requires urgent changes at the level of the components of the three dimensions to reach sustainability values. | Critical |
| (3) | Minimum acceptable threshold of sustainability of quinoa production systems, and the systems need to implement measures to improve their valuation since any adversity in the components of the three dimensions can affect sustainability. | Weak |
| (4) | Medium level of sustainability. Although it is a scale close to the optimal value (5), it is necessary to implement mechanisms for continuous improvement at the economic–technological level, use, and conservation of resources, family, and community. | Well-being |
| (5) | Maximum threshold or high level of sustainability of quinoa production systems to remain at these levels, the production systems need to implement internal quality control mechanisms, efficiency, and effectiveness in using inputs, profitability, and high levels of respect and coexistence in the agroecosystems of the communities of Chiara, Ayacucho. | High |

## 2.7. Validation of Consistency of Indicators

The indicators defined according to the methodology and conceptual framework [36] were used, following the guidelines established by [37,38]. Adjustments were made to the model used by [39]. To ensure ease and low cost in obtaining and interpreting them, 10 indicators and 24 sub-indicators were chosen. These measures provided the necessary information to identify trends in potato and quinoa production and marketing, as well as resource use and dependence on external inputs, among other factors that influence the sustainability of production systems in the area analyzed [36,40].

The participatory rural appraisal methodology was to carry out a complete analysis of the agroecosystem to collect information. Surveys containing structured questions applied to 50 producers selected through simple random sampling were used to ensure that each member of the population had the same probability of being included in the study [41–43]. The comparison of production units was a valuable strategy to address the different dimensions of sustainability, which involved standardizing the data on a scale of 1 to 5, with a value of 5 reflecting the highest degree of sustainability and 1 indicating the lowest level. It was thus possible to combine different indicators covering different dimensions and characteristics of the system. By considering various indicators, such as economic, environmental, and

social, it was possible to obtain a more complete and accurate view of the sustainability of the production system [44].

## 3. Results

### 3.1. Sustainability Assessment

Sustainability is a complex concept with multiple dimensions, as it encompasses economic, ecological, productive, social, and cultural objectives and temporal ones. Therefore, its evaluation is not as straightforward as comparing performance. Indicators must be developed to simplify this complexity and clearly show trends [45]. The variables were defined to evaluate the economic, environmental, and social indicators that were selected based on a literature review, critical aspects related to the activity of organic, conventional, and alternative potato and quinoa production systems, the characteristics of the producers in the region, conditions observed in the field and consultation with professionals in the field of research. In identifying indicators and variables, descriptors and criteria were defined. Indicators were chosen that are easy to obtain and interpret, provide the necessary information and allow for the detection of trends in the study area. These are composed, in turn, of sub-indicators and selected and quantified variables that make up, respectively, the selected indicators or sub-indicators. The results were obtained through surveys applied during the field phase. To elaborate on the indicators, we followed the methodology and the proposed conceptual framework and guidelines [37,38].

### 3.2. Sustainability of Economic Indicator

From the results of the cultivated area and production area, 85.5% of the farmers sow areas between 0.25 and 0.5 ha. Of the farmers, 14.5% have smaller sizes than 0.25 ha (Figure 4a). It can be seen that the plots are small and there are no farmers with areas larger than 1 ha in the Aynoka of the farming community of Palermo Rio Salado. As economic growth is experienced, people's standard of living indeed tends to improve [46].

In the sample, 100.0% of the farmers report a productivity or yield per hectare between 5 and 10 tons/ha. The low products in potato cultivation are due to several negative factors, including poor soil preparation with an animal-drawn plow, which barely scrapes the surface, resulting in a planting depth of only 0.12 cm and a distance between furrows that varies between 0.60 and 0.70 cm; seed potatoes of traditional varieties with reduced weight are used, little natural fertilizer is incorporated, and planting is done outside the optimum date [4]. In addition, cultural work and pest and disease control are deficient, which leads to low production and productivity. As for the selling prices of potatoes in Peru, they can vary according to the season, supply and demand, and other factors such as the region of the country and the type of potato. Potato prices vary between 0.50 PEN and 3.00 PEN per kilogram. Potatoes grown in the Altiplano region have sustainable prices due to local market preference.

Concerning quinoa and potato production, it is an essential source of income for local farmers. However, the quality of the villagers' houses can affect their well-being and quality of life. According to the information gathered through the surveys, 23.6% of the homes are of noble material and excellent quality, 45.5% are of noble material and good quality, and 3.9% are of regular adobe material and earthen floor, according to Figure 4c. According to the current study, which included a survey of 55 producers, it was observed that only 1 of the farmers reached a level of integration considered very good. Another group, representing 25.5% of the respondents, demonstrated integration rated as good. The next group, representing 65.5% of the participants, showed a fair level of integration. Finally, 7.3% of the farmers were identified as poor needing better integration. In Aynoka, the diversity in the supply of products for sale in potato and quinoa (Table 3) ranges between 3 and 5 products. Among these products, potatoes and quinoa are the predominant crops used for family food and marketing. In addition to potato and quinoa, cereals and legumes are grown occasionally, mainly for family consumption, and can also be marketed. Diversification in present-day farming happens via genetics

and landscape changes facilitated by technology [47]. Furthermore, diversification of agricultural production decreases susceptibility to climate change and improves soil quality and fertility, as mentioned in the study by [48].

(a)

| A1. Cultivated surface | Nº People | % | A2. Productivity | Nº People | % | A3. Pest incidence | Nº People | % |
|---|---|---|---|---|---|---|---|---|
| Over 1 ha | 0 | 0.0 | Over 30 t/ha | 0 | 0.0 | Very low | 45 | 81.8 |
| 0.75–1 ha | 0 | 0.0 | 20–30 t/ha | 0 | 0.0 | Low | 10 | 18.2 |
| 0.5–0.75 ha | 0 | 0.0 | 10–20 t/ha | 0 | 0.0 | Medium | 0 | 0.0 |
| 0.25–0.5 ha | 8 | 14.5 | 5–10 t/ha | 55 | 100.0 | High | 0 | 0.0 |
| Less than 0.25 ha | 47 | 85.5 | Less than 5 t/ha | 0 | 0.0 | Very high | 0 | 0.0 |
| **B1. Net income** | **Nº People** | **%** | **C1. Diversification for sale** | **Nº People** | **%** | **C2. Marketing distribution** | **Nº People** | **%** |
| Over 10,000 (S/. /ha) | 0 | 0.0 | Over 5 products | 6 | 10.9 | Local market | 55 | 100.0 |
| 5,000–10,000(S/. /ha) | 9 | 16.4 | 4 products | 26 | 47.3 | National market | 0 | 0.0 |
| 2,500–5,000. (S/. /ha) | 13 | 23.6 | 3 products | 23 | 41.8 | To the collector | 0 | 0.0 |
| 1,000–2,500. (S/. /ha) | 21 | 38.2 | 2 products | 0 | 0.0 | Wholesale | 0 | 0.0 |
| Less than 1,000 (S/. /ha) | 12 | 21.8 | 1 products | 0 | 0.0 | Export | 0 | 0.0 |

| C3. Dependence on inputs | Nº People | % |
|---|---|---|
| He uses all his resources from his farm, he does not buy anything. | 21 | 38.2 |
| He only buys seed, the rest from his own farm | 12 | 21.8 |
| Buy seed, manure, or guano from island | 4 | 7.3 |
| Purchase seed, manure, or island guano and fertilizer | 6 | 10.9 |
| Purchasing (Seeds, fertilizers, pesticides, and others) | 12 | 21.8 |

(b)

| A1. Crop rotation | Nº People | % | A2. Crop diversification | Nº People | % | A3. Organic matter incorporation | Nº People | % |
|---|---|---|---|---|---|---|---|---|
| Every year | 55 | 100.0 | Over 5 | 0 | 0.0 | Over 5t/ha | 4 | 7.3 |
| Twice a year | 0 | 0.0 | 4 | 33 | 60.0 | 3–4.9 t/ha | 13 | 23.6 |
| Every 3 year | 0 | 0.0 | 3 | 22 | 40.0 | 2–2.9 t/ha | 26 | 47.3 |
| Every 4 years | 0 | 0.0 | 2 | 0 | 0.0 | 1–1.9 t/ha | 12 | 21.8 |
| Not | 0 | 0.0 | Only 1 | 0 | 0.0 | Less than 1 t/ha | 0 | 0.0 |
| **A4. Land preparation** | **Nº People** | **%** | **B1. Predominant slope** | **Nº People** | **%** | **B2. Vegetation coverage** | **Nº People** | **%** |
| None-tillage farming | 0 | 0.0 | 0–5% | 55 | 100.0 | Vegetation cover all year | 55 | 100.0 |
| Minimal tillage | 0 | 0.0 | 6–15% | 0 | 0.0 | cover during the production | 0 | 0.0 |
| With a yoke | 0 | 0.0 | 16–30% | 0 | 0.0 | Partial cover | 0 | 0.0 |
| Tractor, harrow one pass | 0 | 0.0 | 31–45% | 0 | 0.0 | Vegetation cover all year | 0 | 0.0 |
| With ploughing, harrowing | 55 | 100.0 | Over 45% | 0 | 0.0 | | | |
| **C1. Conservation in situ of v** | **Nº People** | **%** | **C2. Pest/disease managem** | **Nº People** | **%** | **C3. Quality seed production** | **Nº People** | **%** |
| Over 10 varieties | 5 | 9.1 | Cultural | 49 | 89.1 | Sowing certified seed | 0 | 0.0 |
| 5 - 9 varieties | 5 | 9.1 | Chemical | 6 | 10.9 | Non-certified seed | 13 | 23.6 |
| 2 - 3 varieties | 45 | 81.8 | | | | Selected own seed | 40 | 72.7 |
| 1 varieties | 0 | 0.0 | | | | Seed from local market | 2 | 3.6 |
| None | 0 | 0.0 | | | | Seed of unknown origin | 0 | 0.0 |

(c)

| A1. Housing | Nº People | % | A2. Access to education | Nº People | % | A3. Access to health insurance | Nº People | % |
|---|---|---|---|---|---|---|---|---|
| Very good | 13 | 23.6 | University | 0 | 0.0 | With adequate doctors | 0 | 0.0 |
| Good | 25 | 45.5 | Tecnology | 9 | 16.4 | With moderately equipped | 0 | 0.0 |
| Regular adobe and earth floor | 17 | 30.9 | Secondary | 38 | 69.1 | Poorly equipped and temporary | 49 | 89.1 |
| Unfinished or damaged adobe | 0 | 0.0 | Primary | 8 | 14.5 | Poorly equipped and understaffed | 6 | 10.9 |
| Hut, unfinished deteriorated | 0 | 0.0 | No education | 0 | 0.0 | No personnel | 0 | 0.0 |
| **A4. Services** | **Nº People** | **%** | **B1. Producer satisfaction** | **Nº People** | **%** | **B2. Production systems** | **Nº People** | **%** |
| Water, latrine, electricity | 6 | 10.9 | Very happy | 0 | 0.0 | Traditional | 0 | 0.0 |
| No water, latrine, light | 49 | 89.1 | Happy | 43 | 78.2 | Organic | 33 | 60.0 |
| No water, latrine, no light | 0 | 0.0 | Not fully satisfied | 12 | 21.8 | Mixed cleaner | 22 | 40.0 |
| No water, no latrine, no light | 0 | 0.0 | Dissatisfied | 0 | 0.0 | Conventional | 0 | 0.0 |
| | | | Disillusioned with the life | 0 | 0.0 | | | |
| **C1. Level of social integration** | **Nº People** | **%** | **D1. Level of social assista** | **Nº People** | **%** | | | |
| Very goog | 1 | 1.8 | Ministerio de Agricultura | 49 | 89.1 | | | |
| Good | 14 | 25.5 | None | 6 | 10.9 | | | |
| Regular | 36 | 65.5 | | | | | | |
| Very bad | 4 | 7.3 | | | | | | |
| None | 0 | 0.0 | | | | | | |

Economic indicators  ▮  Environmental indicators  ▮  Social indicators  ▮

**Figure 4.** (**a**) Economic indicators; (**b**) Environmental indicators; (**c**) Social indicators from the results of the surveys carried out among the community members and owners of the Aynoka.

**Table 3.** Indicators of economic sustainability of the potato and quinoa in Palermo Rio Salado, Juli.

|  |  | A |  | B | C |  |  | Potato |
| --- | --- | --- | --- | --- | --- | --- | --- | --- |
| **Economic** | **A1** | **A2** | **A3** | **B1** | **C1** | **C2** | **C3** | **E.K.** |
| Combined | 1.27 | 2.00 | 4.55 | 2.32 | 3.77 | 5.00 | 3.27 | 2.89 |
| Organic | 1.06 | 2.00 | 5.00 | 2.36 | 3.64 | 5.00 | 3.55 | 2.95 |
| Average | 1.17 | 2.00 | 4.77 | 2.34 | 3.70 | 5.00 | 3.41 | 2.92 |
|  |  | A |  | B | C |  |  | Quinoa |
| **Economic** | **A1** | **A2** | **A3** | **B1** | **C1** | **C2** | **C3** | **E.K.** |
| Combined | 1.27 | 3.00 | 4.27 | 2.68 | 3.77 | 3.86 | 4.05 | 3.07 |
| Organic | 1.06 | 3.00 | 4.27 | 2.03 | 3.64 | 4.45 | 3.82 | 2.89 |
| Average | 1.17 | 3.00 | 4.27 | 2.36 | 3.70 | 4.16 | 3.93 | 2.98 |

### 3.3. Sustainability of Environmental Indicators

Crop rotation in the Aynoka used in the Palermo Rio Salado farming community is an ancestral practice that benefits agriculture and the environment. Crop rotation maintains soil fertility, reduces erosion, and prevents the spread of diseases and pests. In this system, farmers grow different crops on the same plot of land for a certain period and then switch to other crops the following planting season. Crop rotation also helps to improve crop quality, as each species absorbs different nutrients from the soil, which helps to maintain soil fertility and reduce reliance on chemical fertilizers. This crop rotation and genetic diversity system is a valuable practice that should be promoted and replicated in other farming communities. Recognizing and valuing ancestral and traditional farming practices is essential, as it can offer sustainable and practical solutions to today's food production challenges and environmental protection.

Another environmental factor is pest control. In the Palermo Rio Salado farming community, 89.1% of the farmers use the cultural control method for pest management, while the remaining 10.9% use the chemical control method. Cultural control refers to agricultural practices that seek to prevent or reduce the appearance of pests, such as using resistant varieties, crop rotation, pruning, and manual elimination of problems, among others. In contrast, chemical control involves the use of chemicals to control pests. While weather events indicate that 71% of agriculture is negatively affected by hail, the remaining 29% is affected by frost. Farmers in the study area lack adequate technologies and techniques to protect their crops from hail. Regarding potato varieties, 84% of the farmers plant between 2 and 3 varieties and 16% between 5 and 9 varieties. The diversity of potato varieties is an essential aspect of traditional agriculture in the Andes because it ensures a more stable and diverse source of food and income. This result shows that within the community, there are still farmers who retain the practice of growing multiple potato varieties and thus reduce the adverse effects of diseases/pests and unpredictable negative changes in climatic conditions. However, it is essential to note that some studies suggest a trend of decreasing crop diversity in the rural areas of the Andes due to urbanization, modernization, and the influence of markets. Similarly, quinoa varieties that 7% of farmers plant two varieties, 40% plant three varieties, and 53% plant up to four types.

In potato cultivation, it was found that variety conservation and seed management played an essential role in the sustainability of the crop. The sub-indicator of variety conservation obtained a value of 3.20, indicating that significant efforts were made to preserve genetic diversity by planting different potato varieties. Ensuring the crop's resilience to diseases and environmental changes is crucial.

On the other hand, the seed management sub-indicator had a value of 3.18, implying that good practices were applied in seed potato management. These include selecting and conserving quality seeds and adopting appropriate storage techniques to ensure long-term viability. According to Table 4, a value of 4.45 was observed for the pest management sub-indicator, indicating that this effective strategy was implemented to control pests

in the potato crop. The primary approach was cultural control, which involves using good agricultural practices such as crop rotation, weeding, and implementing reasonable hygienic procedures in crop management. This has helped minimize the use of chemicals and promoted a more sustainable approach to pest control. Quinoa cultivation has essential aspects related to the conservation of varieties, and seed management was also observed. The sub-indicator to the preservation of quinoa varieties in Table 4 obtained a value of 2.98, which indicates that efforts to preserve the genetic diversity of this plant have been reduced. The conservation of quinoa varieties is crucial to maintain the adaptability and resistance of this crop to various factors, such as climate change and disease.

**Table 4.** Indicators of environmental sustainability of the potato and quinoa in Palermo Rio Salado, Juli.

| | A | | | | B | | C | | | Potato |
|---|---|---|---|---|---|---|---|---|---|---|
| **Environment** | **A1** | **A2** | **A3** | **A4** | **B1** | **B2** | **C1** | **C2** | **C3** | **E.I.** |
| Combined | 5.00 | 3.41 | 3.00 | 1.00 | 5.00 | 5.00 | 3.41 | 3.91 | 3.09 | 3.67 |
| Organic | 5.00 | 3.73 | 3.27 | 1.00 | 5.00 | 5.00 | 3.00 | 5.00 | 3.27 | 3.81 |
| Average | 5.00 | 3.57 | 3.14 | 1.00 | 5.00 | 5.00 | 3.20 | 4.45 | 3.18 | 3.74 |
| | **A** | | | | **B** | | **C** | | | **Quinoa** |
| **Environment** | **A1** | **A2** | **A3** | **A4** | **B1** | **B2** | **C1** | **C2** | **C3** | **E.I.** |
| Combined | 5.00 | 3.41 | 1.00 | 2.00 | 5.00 | 5.00 | 2.95 | 3.73 | 3.32 | 3.98 |
| Organic | 5.00 | 3.73 | 1.00 | 2.00 | 5.00 | 5.00 | 3.00 | 3.91 | 3.67 | 4.09 |
| Average | 5.00 | 3.57 | 1.00 | 2.00 | 5.00 | 5.00 | 2.98 | 3.82 | 3.49 | 4.04 |

On the other hand, the quinoa seed management sub-indicator had a value of 3.49 (Table 4), also suggesting that good seed management practices were applied. This implies selecting and conserving quality seeds and adopting appropriate storage techniques to ensure their viability and quality. Properly managing quinoa seeds ensures good germination, healthy plant growth, and a successful harvest. These findings highlight the importance of variety conservation and seed management in quinoa cultivation. This crop's productivity and resilience are promoted by preserving genetic diversity and using quality seeds. Furthermore, these practices contribute to maintaining the sustainability and continuity of quinoa production, thereby ensuring its availability as a nutritious and sustainable food source. This diversity gives it an innate resilience to abiotic stresses and climate change, allowing it to survive and thrive in challenging environments [49].

*3.4. Sustainability of Social Indicators*

Other results show that 78% of the producers are male and 22% are female. This result is similar to the one reported by [50], in which it is mentioned that out of every 10 agricultural producers, about 7 are men and 3 are women. In this rural community, women are still mainly in charge of traditional tasks related to household care, such as cleaning, taking care of clothes, or preparing meals. At the same time, men manage production and crop management. A similar observation is reported in Mexico [51]. A large part of the producers is between 30 and 60 years old; together, they represent 48.9% of the population. A similar study conducted in Ayacucho found that 82% of producers were between 20 and 49 years old. This population is more susceptible to change and adopting new technological innovations.

In the community of Palermo Rio Salado, 100% of the farmers have some level of education, 14.5% of the respondents have primary education, are the oldest farmers, and generally reside near the Aynoka. Next, 69.1% of farmers have secondary education and in age are below 55 years old and live in the countryside as well as in the vicinity of the city, and 16.4% have a technological level of education due to the availability of this type of education in the setting and also reside in the countryside and the city this. El [52] mentions that younger and more educated farmers can adopt new technologies or production systems more quickly.

During the social sustainability indicator assessment, including mixed and organic production systems, a survey was conducted to obtain relevant information on social sustainability in these crops. This survey made it possible to collect accurate and representative data on different social aspects related to potato and quinoa production, which provided a complete picture of the social sustainability of both crops.

The Social Indicator (SI) shows values above 3 in both systems, indicating sustainability. In the mixed system, the education level sub-indicator has a critical value of 2.86, reflecting the poor educational background due to lack of access to education, which hinders the adoption of technologies (Table 5). The Social Indicator (SI) values for the mixed and organic production systems for potato and quinoa cultivation are 3.27 and 3.50, respectively. These values indicate that they are in the range of weak sustainability, as they are between 3 and 3.99. Although considered sustainable, aspects can still be improved to achieve greater social sustainability in both production systems.

**Table 5.** Indicators of social sustainability of the potato and quinoa in Palermo Rio Salado, Juli.

| | A | | | | B | | C | D | Potato and Quinoa |
|---|---|---|---|---|---|---|---|---|---|
| Social | A1 | A2 | A3 | A4 | B1 | B2 | C1 | D1 | SI |
| Combined | 3.45 | 2.86 | 2.73 | 3.55 | 3.45 | 3.00 | 3.14 | 3.91 | 3.27 |
| Organic | 4.24 | 3.12 | 3.00 | 3.00 | 4.00 | 4.00 | 3.27 | 5.00 | 3.50 |
| Average | 3.85 | 2.99 | 2.86 | 3.27 | 3.73 | 3.50 | 3.20 | 4.45 | 3.38 |

*3.5. Sustainability of the Ancestral Amoeba Diagram Indicators*

The amoeba diagram (Figure 5a,b) shows that the organic system is less dependent on external inputs and pest incidence. Nevertheless, such a system has three critical points: productivity, net income, and cultivated area. However, there are alternatives or trade channels that could have a positive impact on profitability.

One of the highlights of the marketing strategy was the producers' ability to reach the local market directly. This strength translated into many benefits for both producers and consumers. Producers could obtain a higher profit margin by eliminating intermediaries in the marketing channels by selling directly to the final consumer. It allowed them to get a fair price for their work and effort and have greater control over the sales process. Farmers' Markets are places where farmers and artisans sell goods directly to consumers from stalls, highlighting the direct producer–consumer relationship [53].

In the farming community of Palermo Rio Salado, the prevalence of short-distance marketing channels stands out. It has advantages for buyers, as they can purchase potato and quinoa products at a lower cost. The variables of cultivated area, productivity, and net income are far from optimal levels of sustainability, which had a significant impact on the low value of the EK indicator. Furthermore, as shown in Figure 5a,b, the indicators of maximum sustainability are associated with pest incidence and marketing channels. These factors contribute positively to sustainability. However, cultivated areas and marketing channels presented the values furthest from the sustainability threshold, indicating the need for attention and improvement.

From Figure 5c, it can be seen that, in potato cultivation in both the mixed and organic systems, land preparation stands out as a critical indicator due to the intensive use of farm machinery. This result indicates how land preparation significantly impacts potato cultivation, especially regarding machinery management. It is important to consider sustainable practices that minimize the negative impact on the soil and preserve its quality in the long term.

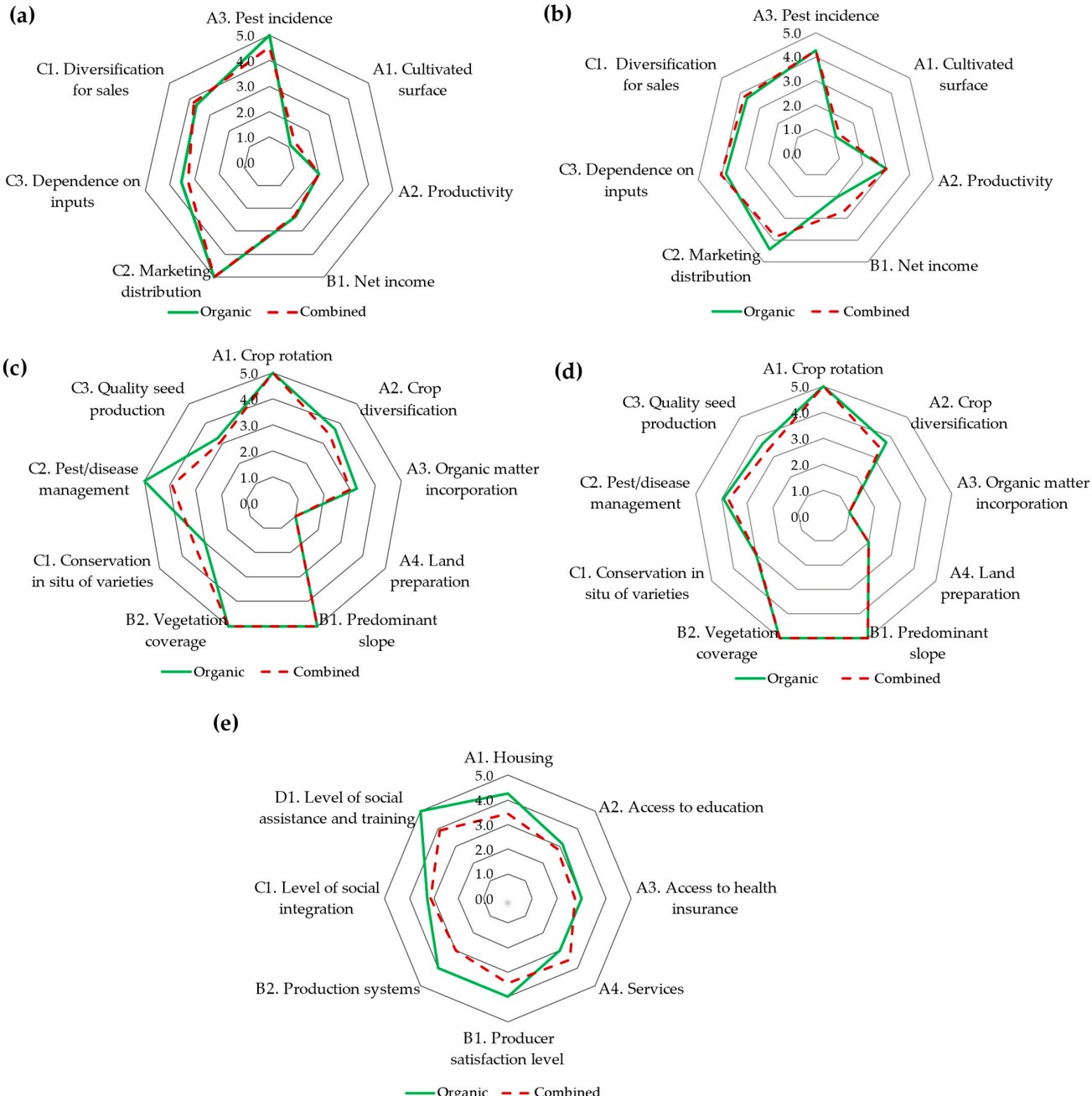

**Figure 5.** Production systems economic dimension (**a**) potato; (**b**) quinoa; production systems environmental dimension (**c**) potato; (**d**) quinoa; production systems social dimension (**e**) potato and quinoa.

On the other hand, Figure 5d shows that in quinoa cultivation, both in the mixed and organic systems, soil preparation and incorporation of organic matter are critical indicators. This is due to quinoa's low soil fertility requirements, significantly when grown after the potato is planted the previous year. Since potato is a more nutrient-demanding crop, it is crucial to ensure adequate soil preparation and sufficient incorporation of organic matter to ensure that quinoa has the right conditions for healthy growth.

These findings highlight the importance of considering crops' land preparation practices and organic matter incorporation. They also point to the extent of crop rotation to maximize soil use efficiency and maintain soil fertility in the long term. It is essential to adopt sustainable approaches that promote soil conservation and crop health, contributing to the sustainability and productivity of potato and quinoa cropping systems.

The value of the Environmental Indicators of organic production is closer to the ideal situation, reaching a score of 4.04. However, it is vital to address a critical aspect related to the indicator of land preparation and conservation of quinoa varieties, whose value of 2.98 reflects the displacement and loss of local quinoa ecotypes due to adopting improved varieties. According to [54], starting in the 1900s, roughly 75 percent of plant genetic diversity disappeared as farmers globally transitioned from diverse landraces to uniform, high-yielding strains. This was supported by [55], who, in a study of quinoa production systems in Chiara, point to a trend of loss of landraces in traditional techniques due to replacement by varieties in high market demand.

According to the data presented Figure 5e, specific indicators are closer to the ideal level of sustainability in the context analyzed. Specifically, housing quality stands out, with a value of 3.85, which implies that a satisfactory grade has been achieved regarding habitability conditions for the producers. In addition, the level of producer satisfaction reaches a value of 3.73, indicating a considerable degree of satisfaction concerning their agricultural activity. Finally, technical assistance obtains the highest value among the indicators evaluated, with an outstanding 4.45, indicating that adequate technical support is provided to producers.

The indicators related to essential services, production systems, and social integration show a weak level of sustainability, as their values barely exceed the minimum threshold. This indicates that there are significant challenges in these areas in terms of their sustainability in the assessed context. In other words, the indicators for essential services, production systems, and social inclusion reflect critical issues concerning these crucial aspects. The values recorded are slightly above the minimum required, but much work still needs to be done to improve and strengthen their sustainability. In summary, the assessment indicates that essential services, production systems, and social inclusion require more attention and additional efforts to achieve a more robust level of sustainability in the context analyzed.

On the other hand, critical points were identified in the health services, where 11% of the producers perceive regular attention with a lack of equipment and unqualified personnel, while 89% qualify as poorly equipped and with temporary personnel. Furthermore, the level of education also appears to be a critical aspect in the evaluated context. Social sustainability implies ensuring that communities and individuals have access to resources, services, and opportunities that enable them to meet their basic needs, develop fully, and enjoy a good quality of life. It is essential to consider different dimensions, such as equity, citizen participation, social justice, cultural diversity, security, and social cohesion [56].

### 3.6. General Sustainability Index (GSI)

Figure 6a shows that the values obtained for the Environmental Sustainability Index (ESI), 3.74, and the Social Sustainability Index (SSI), 3.38, reflect a medium level of sustainability. However, it is essential to highlight that the Economic Sustainability Index (EKI) presents a limited contribution to the sustainability of potato production, being situated in the weak sustainability range with a value of 2.92. The values obtained for the Environmental Sustainability Index (ESI) and the Social Sustainability Index (SSI) in quinoa production indicate a medium level of sustainability. The ESI reaches a value of 4.04, while the SSI shows a value of 3.38. However, it is essential to note that the Economic Sustainability Index (ESI) has a limited contribution to the sustainability of quinoa production, being in the weak range of sustainability with a value of 2.98 (Figure 6a). Studies indicate a potential to increase the share of sustainable agriculture globally in the field by 40–60%. This increase is based on addressing nitrogen deficiencies affecting organic agriculture [8].

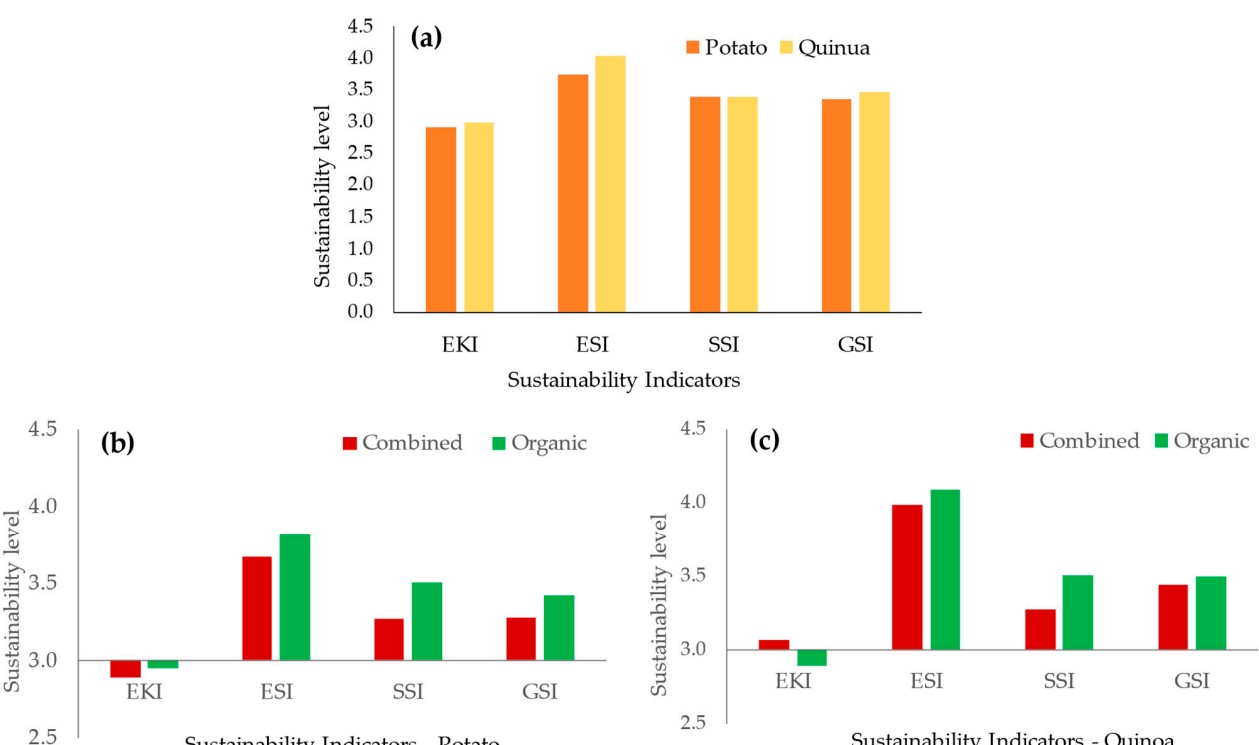

**Figure 6.** (**a**) Overall Sustainability Indicator of potato and quinoa; (**b**) overall Sustainability Indicator of potato; (**c**) overall sustainability of quinoa in the Aynoka.

The results obtained for potato and quinoa cultivation in the mixed and organic production systems, according to the General Sustainability Index (GSI) presented in (Figure 6b,c), indicate medium sustainability. The values of the Environmental and Social Indicators are slightly above the minimum acceptable threshold of sustainability, which implies the need to implement measures to improve their evaluation.

Results reveal that the quinoa boom significantly increased the hectares cultivated with this crop by 43% in 2014, compared to projections without the crash. This phenomenon has led to a rapid expansion of production in ancestral quinoa-growing areas and the introduction of quinoa in new regions. The effects of this growth are already evident in the land cover transitions observed in Peru, including the relocation of other crops, the rebound of quinoa in previously abandoned areas, and the cascading effects along the production chain [57].

## 4. Discussion

### 4.1. Preservation and Sustainability of Potato and Quinoa

Sustainability is a complex, multidimensional concept encompassing economic, ecological, productive, social, and cultural objectives and temporal ones. Therefore, its assessment is more complex than comparing performance. Indicators must be developed to simplify this complexity and clearly show trends [45]. The Altiplano region is characterized by its extensive territory dedicated to potatoes, specifically quinoa cultivation, covering around 37,000 hectares, representing a significant proportion of 44% of national production. Farmers in this region have achieved outstanding yields of over 1200 kg per hectare, demonstrating their commitment and skill in cultivating this highly nutritious cereal.

Agriculture in the highland region of Puno and the farming community of Palermo Rio Salado face severe problems due to the extreme weather conditions that have been occurring frequently in recent years, reducing productivity and crop quality. Other studies indicated that hail is one of the main factors limiting potato and quinoa production in the region, causing crop damage and reducing yields [58]. Adherence to solid sustainability in defining

indicators and sub-indicators demonstrates a deep understanding of the interdependence between agroecosystems and natural resources. By considering natural capital as a provider of indispensable functions, it recognizes the need to preserve and sustainably manage these resources to ensure long-term agricultural development in the farming community of Palermo Rio Salado. It is crucial to address the interrelationship between economic growth, natural resources, and the environment to pursue sustainable development [59].

*4.2. Sustainability Economic, Environmental, and Social Indicators*

Conservation agriculture represents a fundamental arrangement to meet present and future food needs while seeking to improve the livelihoods of farmers and society at large. To achieve this goal, all aspects of agriculture must adhere to sustainable practices [60]. Government support plays a crucial role in achieving sustainable agriculture, as governments can help businesses reduce costs and make it easier for consumers to purchase renewed products [61]. Different approaches to sustainable farming seek to secure food and ecosystem services for both present and future generations amid climate change, higher energy expenses, societal unrest, financial instability, and growing environmental deterioration [60,62,63]. Evaluating sustainability employs indicators to yield quantifiable data, facilitating goal setting through long-term strategies for sustainable development. Sustainable development is the pathway to achieve overall sustainability, emphasizing systemic well-being. In comparison, sustainability embodies the qualitative outcome observed through factors and indices [64].

Potato and quinoa are grown in four production systems: intensive conventional agriculture harms climate change, water pollution, decreased biodiversity, and depletion of natural resources while suffering the direct consequences of these problems [60]; in the traditional production system, low production and productivity rates are recorded in small areas with subsistence economies; mixed or alternative with rational use of agrochemicals and responsible management of the means of production [61]; products grown in the organic system receive technical assistance and are destined for commercialization in international markets [38]. A set of 10 indicators and 24 sub-indicators were identified and grouped according to the three dimensions of sustainable development. The focus was on selecting hands that were of practical use and easy to obtain and interpret, which would allow for the detection of relevant trends in the field of study.

In addition, the presence of varieties from other intensive farming regions can also influence prices, as these varieties can compete with the local potato and reduce their value. This puts small farmers in an even more vulnerable position regarding trade and prices. Lack of collective action and intermediaries dictating low prices are the main constraints to potato marketing in Kenya [65]. This means that smallholder farmers need help to organize themselves to negotiate better prices or to sell directly to the market, resulting in a situation where intermediaries set low prices that do not reflect the actual value of the potato. Decision-makers recognize the importance of economic performance in ensuring the financial sustainability of the primary sector [66]. By protecting and promoting this dimension of sustainability, they seek to balance economic growth, profitability, and the long-term viability of agricultural and related activities. This increase in living standards leads to a greater demand for food to meet the needs of a growing population. More arable land is required to meet this demand [46]. Economic growth and improved living standards lead to an increased demand for food, implying the need to expand arable land and adopt more efficient agricultural practices to ensure long-term food supply.

The study found that most farmers have small cultivated areas ranging from 0.25 to 0.5 hectares, which is common in the region. This is consistent with the subsistence of farming in the area and reflects limited land availability. However, the study also emphasizes the need for increased arable land to meet the growing demand for food due to economic growth and improved living standards. This finding aligns with the understanding that population growth and rising living standards drive increased food demand [28]. Regarding the selling price of quinoa, specifically in the localities adjacent

to the farming community of Palermo Rio Salado, Juli, prices vary from 30 to 50 soles per arroba depending on the variety and the production system, resulting in a cost of approximately 4 soles per kilogram of quinoa grain on the local market. The most important economic factors are the worldwide recognition of quinoa, which led to a price increase between 2013 and 2015 due to high national and international demand. Farmers point to the significant rise in the areas of two high-order varieties in the market: Blanca de Juli and Instituto Nacional de Investigación Agraria (INIA). This reduced the characteristic diversity of the Aynoka, standardizing the fields and thus favoring genetic vulnerability. Although this was positive for the family economy, the country, and the region, it can negatively impact the environment and lead to changes in the farmer's cropping and feeding patterns [67].

The value of the Environmental Indicators of organic production is closer to the ideal situation, reaching a score of 4.04. However, it is crucial to address a critical aspect related to the indicator of land preparation and conservation of quinoa varieties, whose value of 2.98 reflects the displacement and loss of local quinoa ecotypes due to adopting improved varieties. According to [54], starting in the 1900s, roughly 75 percent of plant genetic diversity disappeared as farmers globally transitioned from diverse landraces to uniform, high-yielding strains. This is supported by [55], who, in a study of quinoa production systems in Chiara, point to a trend of loss of landraces in traditional techniques due to replacement by varieties in high market demand. Climate variables support the idea that a one-degree Celsius increase in average temperature causes economic losses of 320 USD per hectare for farmers in the Puno region. However, it has been found that, with climate change adaptation measures, these financial losses could be reduced by 43.9%, even in the most challenging climate scenario [68,69].

The social aspects of sustainability are often mentioned but rarely examined in depth. This pillar is considered the weakest and least explored [70]. After a brief review of existing concepts and theories, this research adopts a case study technique to comprehensively analyze the third pillar of sustainability. It proposes social capital as a measure of social sustainability. To explore the social dimension in crop production, such as potatoes and quinoa, consider various aspects involving communities and related social actors. The Social Indicator (SI) values for the mixed and organic production systems for potato and quinoa cultivation are 3.27 and 3.50, respectively. These values indicate that they are in the range of weak sustainability, as they are between 3 and 3.99. Although considered sustainable, aspects can still be improved to achieve greater social sustainability in both production systems. Some studies concluded that future agricultural and rural development policies should prioritize the social dimension. Consideration should be given to the fact that greater involvement of farmers in improving the economic situation of their families can reduce stress and workload [71]. In Palermo Rio Salado, all farmers possess education to varying extents. Of these, 15% have primary education, typically older and residing close to Aynoka. Next, 69% have secondary education, are generally under 55, living in rural and urban areas. Furthermore, 16% hold technological education available locally. Some authors note that educated and younger farmers adeptly adopt new technologies [52].

On the presence of associations or organizations in Colonche and Manglaralto, Ecuador [72], most farmers surveyed (70%) are affiliated with some association, indicating a social relationship ranging from fair to very good. These results confirm farmers' importance and active participation in associations as a form of collaboration and mutual support in these communities. Traditional gender roles are observed, with women primarily responsible for household tasks and men for production. This reflects cultural norms and is consistent with findings from previous studies [73]. Encouraging the blend of urban and rural development is a powerful approach to tackling regional economic disparities and lower income inequality. By linking and coordinating development efforts in urban and rural areas, positive synergies can be generated that drive economic progress more equitably and sustainably [74]. Urban–rural integration has become an essential issue in sustainable urban development. Additional findings reveal that 78% of the producers are male, while

22% are female. This outcome resonates with the results reported in [50,75], indicating that among every 10 agricultural producers, approximately 7 are men and 3 are women. Within this rural community, traditional gender roles persist, with women primarily responsible for household-related tasks such as cleaning, laundry, and meal preparation. A significant portion of the producers falls within the age range of 30 to 60 years, collectively constituting 48.87% of the population. A parallel study conducted in Ayacucho yielded analogous results, indicating that 82% of producers were 20 to 49 years old.

In Bolivia, the intensifying of quinoa productivity has raised concerns about soil degradation and socio-ecological challenges in addressing climate change and ensuring food security. The Peruvian situation aligns with this: the country has led quinoa production and export since 2014, contributing about 60% of the worldwide yield [57,76]. Ahead, agricultural and rural development policies need to emphasize the social dimension as a top concern. Consideration should be given to the fact that greater involvement of farmers in improving the economic situation of their families can reduce stress and workload [71].

## 5. Conclusions

In the community of Palermo Rio Salado, the Aynoka still preserves many ancestral traditions established by its members. This involves following rotation cycles as part of a rational management strategy to maintain soil management. Although there is a tendency to reduce the diversity of cultivated species, each producer or family uses particular techniques, procedures, measurements, formulations, and weights. In addition, the processing of products such as quinoa (threshing, venting, etc.) and potatoes (storage, etc.) is done at the family level. As the urbanization of the city of Juli expands, the Aynoka will experience changes in their configuration as homogeneous areas of cultivation. These areas will transform into spaces with heterogeneous crops and diverse agricultural facilities. The Palermo Rio Salado Aynoka still maintains the ancestral rules established by the community members. The production of potatoes and quinoa in the Aynoka contributes to a healthier diet, to the economy of the families, and to the food security of the members of the Aynoka.

The analysis of the sustainability of the economic indicator in the potato and quinoa crops of the farming community of Palermo Rio Salado reveals that the minimum sustainability threshold was not exceeded due to the practice of sequential crop rotation, where potato is grown first and then followed by quinoa. Sequential crop rotation involves consecutively growing different crops on the same piece of land. In this case, potato is grown before quinoa. However, the results indicate that this sequence has not achieved satisfactory economic sustainability. It was observed that the potato crop occupies the first place in the crop rotation due to its higher nutritional requirements. This crop is enriched by applying sheep manure, mainly during the first cycle of Aynoka. This practice ensures that nutrients are available throughout the Aynoka and are distributed among the crops being rotated. In the mixed production system, a minimal dose of chemicals is used preventively in the presence of minimal pests, usually caused mainly by summers and excess rainfall.

This approach seeks to control pests early and minimize their impact on crops. These production systems reflect the strategies used by farmers in the community to ensure the success and productivity of quinoa and potato crops. The organic approach is based on the use of resourceful elements and the promotion of soil fertility. At the same time, the mixed system combines the use of chemicals in minimal quantities with preventive practices. These findings highlight the importance of considering crops' land preparation practices and organic matter incorporation. They also point to crop rotation's importance in maximizing soil use efficiency and maintaining soil fertility in the long term. It is essential to adopt sustainable approaches that promote soil conservation and crop health, contributing to the sustainability and productivity of potato and quinoa cropping systems, and in the community of Palermo Rio Salado, the Aynoka still preserve many ancestral traditions established by their members. Finally, the findings underscore the necessity of adopting sustainable practices that foster soil conservation, promote crop health, and contribute to the overall sustainability and productivity of potato and quinoa cropping

systems. Ultimately, the cherished ancestral traditions upheld within the community of Palermo Rio Salado serve as a beacon, guiding the ongoing journey towards sustainable agricultural practices and holistic well-being.

**Author Contributions:** Conceptualization, F.C., L.G., J.Z. and C.C.; Investigation, F.C., L.G., J.Z. and C.M.; Methodology, F.C., L.G., C.M. and E.C.; Formal analysis, F.C., L.G. and M.P.; Data curation, F.C., L.G., J.Z., M.P. and E.C.; Validation, F.C., L.G., M.P. and C.M.; Project administration, F.C., L.G., C.M. and E.C.; Visualization, F.C., L.G., J.Z., M.P., C.C. and E.C.; Resources, F.C., J.Z., M.P., C.C. and E.C.; Software, F.C., C.C., and E.C.; Supervision, F.C., L.G., C.M., and E.C.; Funding acquisition, F.C., L.G., J.Z., M.P., C.C., C.M. and E.C.; Writing—original draft, F.C., L.G., C.M. and E.C.; Writing—review and editing, F.C., L.G., J.Z., M.P., C.C., C.M. and E.C. All authors have read and agreed to the published version of the manuscript.

**Funding:** This investigation was supported by Universidad Nacional del Altiplano—Puno (UNA-PUNO).

**Institutional Review Board Statement:** Not applicable.

**Informed Consent Statement:** Not applicable.

**Data Availability Statement:** Not applicable.

**Acknowledgments:** The authors acknowledge and appreciate the support of the Universidad Nacional del Altiplano—Puno (UNA-PUNO) and the Universidad Nacional Agraria—La Molina (UNALM—LIMA) in their work. Recognizing the institutions contributing to their research or project demonstrates gratitude and a sense of community. Furthermore, it is touching to note that the authors have expressed their appreciation of the primary author's father, Benigno Gregorio Calizaya Ticona, who unfortunately passed away on 26 January 2020. Such dedications carry emotional significance and honor the memory of those who have played essential roles in the authors' lives.

**Conflicts of Interest:** The authors declare no conflict of interest.

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
