# Peer review of "Unveiling Ancestral Sustainability: A Comprehensive Study of Economic, Environmental, and Social Factors in Potato and Quinoa Cultivation in the Highland Aynokas of Puno, Peru"

_sustainability, doi:10.3390/su151713163_

Round 1
Reviewer 1 Report
I found your paper interesting, although I admit I have had limited exposure to the use of indexes. My general concern is with the construction of indexes and their weightings. But this is my concern generally and not specific to your paper. However, you should perhaps note that using different weights (objective or subjective) could alter the results. Even so, you were able to identifiy specific elements in the production practices etc that should be targeted for policy.
I was not sure of how you selected the divisor in your index.
You state that you have chosen the case approach and I think you did a pretty good job since with little knowledge of the subject area and practices I learned a few things.
The paper is well written and the progression is pretty good and the results are nicely summarized.
Reviewer 2 Report
1. What is the main question addressed by the research? In this paper, the authors comprehensively studied the farming mode, sustainability indicators and their impact on potato and quinoa crops in the Aynokas farming system. This study wants to contribute to the development of strategies that increase the sustainability and long-term viability of potato and quinoa production, starting from the specific context of the community of Palermo Rio Salado, Peru. 2. Do you consider the topic original or relevant in the field? Does it address a specific gap in the field? The subject is quite interesting and relevant in the field of sustainability of agriculture and rural economy. This study adds to the knowledge in the field of sustainable agriculture and can serve as an example for other similar areas. 3. What does it add to the subject area compared with other published material? Such studies have been done before (”Sustainability Assessment of Agricultural Systems in Paraguay: A Comparative Study Using FAO’s SAFA Framework”, ”Sustainability Assessment of Smallholder Agroforestry Indigenous Farming in the Amazon: A Case Study of Ecuadorian Kichwas”, ”Review of Existing Sustainability Assessment Methods for Malaysian Palm Oil Production” and others), but from Peru and, especially about potato and quinoa, I haven't seen such studies. 4. What specific improvements should the authors consider regarding the methodology? What further controls should be considered? I consider that the methodology applied by the authors in this work is appropriate and the article can be published with the small editing corrections that I have shown. 5. Are the conclusions consistent with the evidence and arguments presented and do they address the main question posed? In the area of Palermo Rio Saldo Aynoka the ancestral tradition is maintained by the members of the community. Potato and quinoa production in Aynoka contributes to the healthy diet of local people, the family economy and the food security of Aynoka members. Sometimes, in summers with heavy rains, minimal doses of pesticides are used, preventively, to reduce the damage caused by diseases and pests. Crop rotation is essential in maximizing land use efficiency and maintaining long-term soil fertility. In my opinion, the conclusions are consistent with the arguments presented and respond to the purpose of the study. 6. Are the references appropriate? The cited works are numerous, current and appropriate to the topic addressed. 7. Please include any additional comments on the tables and figures. The presented tables and figures are relevant, eloquent and synthetic. They correspond qualitatively to such a work. Minor drafting errors: line 91: wrong ”tions[17]” correct: tions[17]. line 391: wrong ”indicating that This effective”, correct: indicating that. This effective lines 412, 447: wrong ”Palermo rio salado”, correct: Palermo Rio Salado line 681: wrong ”venting, etc.)”, correct: venting etc.). wrong ”storage, etc.” correct: storage etc.Author Response
Please see the attachment.

Reviewer 3 Report
Please see the attached file.

Reviewer 4 Report

Moderate English changes required.
Round 2
Reviewer 3 Report
The authors' explanations and the introduced corrections allow for publication in the current version.